# Enhancing the Biological Activities of Food Protein-Derived Peptides Using Non-Thermal Technologies: A Review

**DOI:** 10.3390/foods11131823

**Published:** 2022-06-21

**Authors:** Gbemisola J. Fadimu, Thao T. Le, Harsharn Gill, Asgar Farahnaky, Oladipupo Odunayo Olatunde, Tuyen Truong

**Affiliations:** 1School of Science, RMIT University, Melbourne, VIC 3083, Australia; fadimugbemisola@gmail.com (G.J.F.); harsharn.gill@rmit.edu.au (H.G.); asgar.farahnaky@rmit.edu.au (A.F.); 2Department of Food and Microbiology, Auckland University of Technology, Private Bag 92006, Auckland 1142, New Zealand; thao.le@aut.ac.nz; 3Department of Food and Human Nutritional Sciences, Faculty of Agricultural and Food Sciences, University of Manitoba, Winnipeg, MB R3T 2N2, Canada; oladipupo.olatunde177@gmail.com

**Keywords:** non-thermal technology, peptides, biological activity, protein modification, hydrolysis, ultrasonication, high-pressure processing, pulse electric field

## Abstract

Bioactive peptides (BPs) derived from animal and plant proteins are important food functional ingredients with many promising health-promoting properties. In the food industry, enzymatic hydrolysis is the most common technique employed for the liberation of BPs from proteins in which conventional heat treatment is used as pre-treatment to enhance hydrolytic action. In recent years, application of non-thermal food processing technologies such as ultrasound (US), high-pressure processing (HPP), and pulsed electric field (PEF) as pre-treatment methods has gained considerable research attention owing to the enhancement in yield and bioactivity of resulting peptides. This review provides an overview of bioactivities of peptides obtained from animal and plant proteins and an insight into the impact of US, HPP, and PEF as non-thermal treatment prior to enzymolysis on the generation of food-derived BPs and resulting bioactivities. US, HPP, and PEF were reported to improve antioxidant, angiotensin-converting enzyme (ACE)-inhibitory, antimicrobial, and antidiabetic properties of the food-derived BPs. The primary modes of action are due to conformational changes of food proteins caused by US, HPP, and PEF, improving the susceptibility of proteins to protease cleavage and subsequent proteolysis. However, the use of other non-thermal techniques such as cold plasma, radiofrequency electric field, dense phase carbon dioxide, and oscillating magnetic fields has not been examined in the generation of BPs from food proteins.

## 1. Introduction

Bioactive peptides (BPs) are protein fragments that have a positive impact on body functions [1]. They play an essential role in the metabolic functions of human health, and over 1500 BPs with various health-promoting properties have been isolated and documented [2]. BPs can be incorporated directly into different foods or encapsulated using biodegradable polymers for improved bioavailability and stability [3]. BPs have been documented to possess several novel activities, including antidiabetic, antihypertensive, antimicrobial, antiviral, antioxidative, immunomodulatory, opioid, and antithrombotic properties [4]. As a panacea to serious health problems due to antibiotic-resistant bacteria, BPs have evolved as a potential candidate for inhibiting bacterial proliferation, which has received a considerably large amount of research interest. Nevertheless, the production and bioactivity of BPs from different food sources depend on many factors including the source, amino acid composition, molecular weight, and most especially method of production. There has been a keen interest in the production of hydrolysates containing BPs for application in functional foods that promote health. Hence, the production of BPs with improved bioactivity and yield has become of the utmost importance to researchers. Milk, particularly bovine milk, is one of the most studied sources for the generation of BPs owing to its relative abundance, uniqueness, and several health-promoting properties of the peptides derived from its proteins [5,6]. Apart from bovine milk, proteins from other sources such as legumes [7], seafood [8], egg [9], beef muscle [10], chicken [11], camel milk [12], walnut [13], watermelon peel and seed [14], and buffalo milk [15] have been successfully used to produce BPs.

BPs can be isolated from a wide range of the aforementioned raw materials using different techniques. Generally, BPs are inactive when present in their parent proteins but become active when cleaved by chemical hydrolysis [16], in vitro enzymatic hydrolysis [17], fermentation with lactic acid bacteria [18], or DNA recombinant technology [19]. The mechanisms of action of these BP extraction or production techniques have been extensively reviewed by Daliri et al. [20]. Among these techniques, enzymatic proteolysis has several advantages such as minimal damage to the nutritional value of protein, low cost of production, process repeatability, and reproducibility, as well as flexibility in upscaling when compared to other preparation methods [21]. Hence, enzymatic hydrolysis has become the most used production method for BPs. Different enzymes (alcalase, savinase, flavourzyme, neutrase, trypsin, pepsin, and papain) have been employed to liberate BPs from food proteins [22,23]. For most proteins, the active site, a region where the enzyme binds with the substrate to catalyze the reaction, is markedly influenced by the protein conformation [24]. This, in turn, affects the efficacy of the enzymes in hydrolyzing the protein. Furthermore, the activities of the indigenous proteases in food protein contribute greatly to the yield and properties of the BPs generated [9,25]. Therefore, the pre-treatment of protein prior to enzymatic hydrolysis, either for changing protein conformation or inactivating indigenous proteases, has become essential to circumvent these limitations [26].

Conventionally, thermal processing has been employed for pre-treating proteins prior to proteolysis. Food proteins are pre-heated prior to hydrolysis to inactivate enzyme-producing microorganisms as well as indigenous enzymes, which may interfere with the hydrolysis process [3]. Additionally, food proteins are also denatured to expose the active site for enzymolysis [25]. According to Gauthier and Pouliot [27], the key factors influencing the diversity of peptides produced during hydrolysis include the protein source, specificity, and quality of enzyme used, as well as the pre-treatment methods. However, heat treatment has been associated with causing protein coagulation, resulting in less exposure or less availability of peptide bonds for enzymic cleavage [9]. Therefore, attention has been drawn to non-thermal treatments such as ultrasound, pulsed electric field, and high-pressure processing for the pre-treatment of food protein proteins prior to enzymolysis.

Ultrasonication is a technique that utilizes the frequency of sound waves above 20 kHz; a frequency higher than the detection level of human audition [3]. Ultrasound generally acts by generating acoustic cavitation in the biological matrix, and it has been found to have various effects on the bioactivity and functionality of food proteins. On the other hand, HPP, which is referred to as high hydrostatic pressure processing or ultra-high pressure (UHP), involves the exposure of food materials to pressure (100 to 1000 MPa) where instant and even transmissions of pressure throughout the sample allow inactivation of microorganism without significant changes in quality attributes and nutritional components. It modifies protein secondary structure via hydrogen ion and electrostatic interactions, the tertiary structure via hydrophobic and hydrogen bonding, as well as the quaternary structure through hydrophobic interactions [28]. PEF is based on electroporation and cell disintegration by applying repeated pulses when the food is placed between two parallel electrodes. The electromagnetic wave and electric fields generated during PEF treatment can modify the native structure of protein for the release of health-promoting peptides. All these methods have great potential in modifying protein structures, thereby exposing its active sites, which were originally hidden to proteolytic enzymes, thereby enhancing the production of BPs.

Effects of non-thermal pre-treatments on the bioactivity of peptides derived from different food proteins have been the focus of many studies. Garcia-Mora et al. [28] observed that high-pressure treatment (100–300 MPa, 15 min, 40 °C) prior to proteolysis led to complete degradation of lentil proteins which increased the yield and bioactivities of the resulting BPs. A similar study carried out on peanut protein using ultrasonic-assisted enzymolysis revealed a significant improvement in antioxidant activity of the BPs compared to untreated counterparts [29]. More recently, the impact of ultrasound as a pre-treatment in the preparation of peptide-rich hydrolysate from lupin protein revealed that application of ultrasound prior to enzymolysis with alcalase and flavourzyme improved the antioxidant, antihypertensive, and antidiabetic properties of lupin protein hydrolysates [30,31,32]. The influence of ultrasound pre-treatment on the bioactivities of BPs derived from whey protein [33], duck albumen [9], wheat gluten [34], α-lactalbumin, and β-lactoglobulin [35] have also been documented. However, the underlying mechanisms behind these technologies have not been fully elucidated. Thus, the utilization of non-thermal technologies to maximize the yield, bioactivities, and functionality of BPs derived from food protein is still a growing area of research. Therefore, this review aims to provide an overview of the effect of non-thermal processing technologies as a pre-treatment method prior to enzymolysis production of BPs from different food proteins. In addition, the opportunities for improving the bioactivities of these food-derived BPs in relation to non-thermal food processing technologies are also discussed.

## 2. Production and Bioactivities of BPs by Enzymatic Hydrolysis

As aforementioned, several studies have shown that enzyme hydrolysis is the most suitable method for producing BPs from food proteins [3,20,26,36]. Enzymatic hydrolysis of food proteins is the process of breaking down food proteins by the action of proteolytic enzymes (crude or purified) at a given temperature and pH, intending to apply the hydrolysate or hydrolyzed protein as a food ingredient. Proteins may be hydrolyzed with one or more proteolytic enzymes to produce a peptide-rich hydrolysate (Table 1). Enzymes used in the hydrolysis of food proteins can be classified according to their origin (plant, microbial, or animal), catalytic action (exopeptidase or endopeptidase), and the nature of the catalytic site (e.g., serine proteases, aspartic proteases, metalloproteases, cysteine proteases, and threonine proteases) [37]. Exopeptidases, on the other hand, hydrolyze proteins through cleavage of the terminal peptide bond, releasing dipeptides or single amino acids [38]. Endopeptidases hydrolyze proteins by breaking peptide bonds of non-terminal amino acids, i.e., within the protein molecule. They are the preferred enzymes for hydrolysis of food proteins due to their ability to yield low molecular weight peptides [39], which have been known to possess strong antioxidant and antihypertensive activities [40]. Proteases with the potential to yield low molecular weight peptides are useful for commercial production of antihypertensive and antioxidant peptides [41]. However, the BPs produced with endopeptidases are usually bitter due to the exposure of the hydrophobic amino acids during hydrolysis. Hydrophobic groups of amino acids responsible for bitterness are tryptophan, phenylalanine, isoleucine, threonine, valine, and leucine [8]. The application of BPs has been limited due to the intense bitterness. Several debittering methods have been applied to minimize this adverse effect on human sensorial perception. For instance, removal with alcohol, activated carbon treatment, Maillard reaction, encapsulation, the use of cyclodextrin, chromatographic separation, plastering reaction, and further enzymatic hydrolysis using exopeptidases have been developed.

The isolation of protein from its matrix to remove non-protein compounds, which may interfere with the process, is usually the first step in enzymatic hydrolysis. During enzymolysis, protons are released because of breakage of peptide bonds. This may cause fluctuations in the pH of the medium, which, if not adequately monitored, may affect the hydrolytic efficiency of enzymes, that is, the degree of hydrolysis [26]. The pH must be controlled by the addition of alkali or acid to maintain a high level of enzyme activity at the enzyme optimum pH [55]. The type of peptides generated in protein hydrolysates depends on the type of enzyme used, temperature, and time of hydrolysis [36]. These factors also affect the degree of hydrolysis (DH), which is the ratio of the number of the cleaved peptide bonds to the total peptide bonds [56]. DH is an important factor in the enzymatic hydrolysis of proteins. The influence of these key parameters in enzymatic hydrolysis of food proteins was extensively reviewed by Daliri, Oh, and Lee [20].

## 3. Bioactivities of BPs Derived from Food Proteins Using Enzymolysis

BPs have shown promising potential in performing several beneficial functions in the body. Figure 1 shows the various biological activities possessed by BPs generated from numerous food sources. Typically, BPs have 3 to 20 amino acid residues whose amino acid composition and sequence govern their biological activities [57]. For example, the antioxidant activity of peptides has been linked to the presence of amino acids such as lysine, valine, tyrosine, alanine, histidine, leucine, methionine, proline, cysteine, and tryptophan [58], while valine–alanine–proline epitope has been specifically linked to strong ACE-inhibitory activity [59]. The molecular weight (MW) of BPs also contributes greatly to their bioactivities, in which BPs derived from food protein with an MW below 10 kDa have been reported to possess potent biological activities [60], especially antioxidant and antihypertensive properties [41,61,62]. Hence, the production of BPs with high biological activities requires careful selection of proteolytic enzymes in combination with appropriate pre-treatment that could yield low MW peptides. The following sections focus on the mechanisms of well-documented bioactivities of food protein-derived BPs such as ACE-inhibitory, antidiabetic, antioxidant, antiproliferative, and antimicrobial properties.

### 3.1. Antioxidant Properties of BPs

Increased endogenous antioxidant defense mechanisms in conjunction with augmented production of reactive oxygen species (ROS) in the human body could cause oxidative stress, which is a major contributing factor for the development of several vascular diseases [63]. Biological macromolecules such as lipids, proteins, and DNA are easily damaged by ROS. Explicitly, low-density lipoprotein (LDL) could be modified by oxidation, resulting in increased atherogenicity of oxidized LDL. Therefore, development of severe tissue injury could be attributed to prolonged ROS production [64]. Synthetic antioxidants are often used to prevent the harmful effects of free radicals in the body. However, studies have shown that synthetic antioxidants could be toxic and unsafe for the body [65]. As such, natural antioxidants have been sourced from different food materials due to antioxidant potentials with few or no side effects. BPs are also known to possess antioxidant activity, showing excellent metal ion (Fe^2+^/Cu^2+^) chelating potential and the ability to inhibit oxidation, creating a niche for their applications as a natural antioxidant [66].

Free radicals or ROS with high energy, especially hydroxyl radicals, can interact with all 20 amino acids. However, the imidazole-containing amino acid (His), aromatic amino acids (Tyr, Phe, and Trp), and nucleophilic sulfur-containing amino acids (Met and Cys) demonstrated the most reactivity [67]. Non-enzymatic and enzymatic (lipoxygenase-mediated) peroxidation of essential fatty acids and lipids has been retarded by the different BPs derived from food [68,69]. Singlet oxygen quenching, free radical scavenging, and metal ion chelation are possible mechanisms of antioxidant properties of these peptides. Several studies have documented the antioxidative activities of BPs generated from plant- and animal-derived protein, including wheat gluten [34], peanut [29], poppy seeds and oils [68], prickly pear [69], lentil [28], and camel milk [70]. These studies demonstrate that protein isolation method, type of enzymes used, hydrophobicity, degree of hydrolysis, amino acid composition, concentration, and position of the peptide in the protein structure determine the antioxidant properties of BPs. Zheng, Li, and Li [23] reported that BPs prepared from coconut cake albumin using alcalase had higher antioxidant properties when compared to those prepared using flavourzyme, pepsin, and trypsin. Antioxidant activity was found to be enzyme/substrate-dependent as BPs prepared from kafirin using alcalase, when 0.2 Au/g enzyme-to-substrate ratio was used, had higher activities [71]. Recent advances in the antioxidant properties of BPs produced from different food proteins are shown in Table 2.

### 3.2. ACE-Inhibitory Properties of BPs

Hypertension is one of the leading chronic diseases in the world today. It is characterized by high systolic blood pressure value above 140 mmHg and diastolic pressure above 90 mmHg (140/90) [36]. ACE plays an essential role in the regulation of blood pressure through different reactions in the renin–angiotensin–aldosterone system (RAAS) and kinin nitric oxide system (KNOS). However, RAAS has gained more research attention than other physiological mechanisms underlying hypertension [36]. The main enzymes involved in the RAAS system are renin and ACE [79]. Given that ACE is one of the primary regulators of blood pressure, the inhibition of this enzyme is considered as one of the best strategies for the treatment of hypertension [80]. In the human system, the regulation of blood pressure is critically controlled by ACE, which promotes the conversion of angiotensin I to the potent vasoconstrictor angiotensin II and inactivates the vasodilator bradykinin (Figure 2). ACE inhibitors could inhibit these processes; thus, they can be used as antihypertensive agents [63]. Recently, attention has been directed towards ACE-inhibitory peptides from food sources since synthetic antihypertensive drugs have been associated with several side effects, including cough, headache, renal problems, dysgeusia, and dizziness [81].

Proteins from different foods have been demonstrated to be a promising source of ACE-inhibiting peptides [82,83,84]. Table 3 shows the range of ACE-inhibitory peptides derived from different food proteins such as bovine milk proteins, non-bovine milk proteins, fish proteins, egg proteins, and plant-based proteins. These BPs have created a niche in both the treatment and prevention of hypertension [84,85]. For a BP to exert an antihypertensive effect, the peptide ingested must have resistance to further breakdown or degradation by gastrointestinal tract (GIT) enzymes, which will enable these peptides to reach target sites in an active form [86]. Hence, ACE inhibition potential is evaluated in vitro under GI conditions and expressed by the 50% inhibitory concentration (IC_50_) value. BPs (Arg–Val–Cys–Leu–Pro) obtained from lizard fish muscle protein using neutral protease had ACE-inhibitory activity with an IC_50_ value of 175 µM [87]. Hydrolysate obtained from barbel muscle protein by digesting the muscle with alcalase at a ratio of 1:3 protein/enzyme (mg/U) for 5 h shown high ACE inhibitory activity (IC_50_ = 0.92 mg mL^−1^) [88]. It is documented that several ACE-inhibitory peptides (IVPN, FPGPIPK, QPPQ, IPPK) can be generated from the proteolysis of buffalo milk protein [15]. In a related study, ACE-inhibitory peptides (PEQSLACQCL, ARHPHPHLSFM, QSLVYPFTGPI) with inhibitory activities (IC_50_ value: 4.45 µM) comparable to that of a well-known antihypertensive drug, captopril (IC_50_ value: 4.27 µM), were generated from goat milk casein and whey proteins hydrolyzed with gastric pepsin [89]. In the study of Darewicz et al. [51], seven potent ACE-inhibitory peptides (IVY, VW, TVY, VPW, IW, ALPHA, and IWHHT) were isolated from salmon protein hydrolysate prepared using corolase PP.

The influences of proteolytic enzymes on the ACE-inhibitory properties of BPs have been documented. Lee et al. [99] reported that ACE-inhibitory activity differs between BPs obtained from tuna frame protein using different enzymes such as pepsin, alcalase, papain, neutrase, trypsin, and α-chymotrypsin, in which BPs obtained with pepsin demonstrated the highest activity. This was attributed to the individual characteristics of the BPs generated using the aforementioned enzymes. BPs generated from rohu roe proteins using pepsin had higher ACE-inhibitory activity (47%) than that obtained with trypsin (36%) when the same concentration (mg/mL) was used [100].

The impact of the molecular weight of BPs on their ACE-inhibitory properties has also been demonstrated. Lee, Qian, and Kim [99] reported that the MW of BPs obtained from tuna frame using pepsin influenced the ACE-inhibitory activity, in which the 1–5 kDa fraction showed the highest activity as compared to <1 and 5–10 kDa fractions. Overall, the BPs with MW of 2482 Da and amino acid sequence (Gly–Asp–Leu–Gly–Lys–Thr–Thr–Thr–Val–Ser–Asn–Trp–Ser–Pro–Pro–Lys–Try–Lys–Asp–Thr–Pro) had the highest ACE-inhibitory properties (IC_50_ = 11.28 μm) [99]. When pepsin and trypsin (2% enzyme concentration each) were used for the preparation of ACE-inhibitory peptides from fermented bovine milk, peptide fractions below 3 kDa exhibited the highest ACE-inhibitory activity [101].

In addition to in vitro analysis under GI conditions, spontaneously hypertensive rats (SHRs) are used to study the antihypertensive effect of BPs in vivo [102]. Geerlings et al. [92] reported the identification of three novel ACE-inhibitory peptides (SLPQ, TGPIPN, SQPR) from goat milk BPs. The effectiveness of these three peptides was tested using the SHR model system. It was shown that SHRs fed diets containing peptides for 12 weeks had lower (about 15 mmHg) systolic blood pressure compared to those fed with the control diet. In addition, salmon fish-derived peptides tended to possess a substantial ACE-inhibitory property.

Several peptides with an ACE-inhibitory property have also been generated from plant proteins such as rice bran [91], sesame [53], lentil [28], corn [72], and mung bean proteins [49]. The effectiveness of ACE-inhibitory peptides from plant proteins in blood pressure management using SHRs has also been documented. In a study by Girgih et al. [44], a strong correlation was established between synthetic drug and hemp seed protein-derived peptides in their ability to lower blood pressure after oral administration of the BPs to SHRs. In another study, a novel ACE-inhibitory peptide from bitter melon seed proteins was found to have a blood pressure-lowering effect at a concentration of 2 mg/kg body weight in SHRs [103].

### 3.3. Antidiabetic Properties of BPs

Diabetes is a chronic disease characterized by an elevated blood sugar level above the normal level due to inadequate secretion of insulin or failure in production of insulin [104]. It affects 463 million individuals globally, and over 10% of global health expenditure was spent on diabetes in 2019 (IDF, 2019). Diabetes may be classified into type I, type II, and gestational diabetes, depending on the underlying cause of the disease [104,105]. Globally, type II diabetes is the most common form of diabetes, with 90% of all cases reported as this type [106]. Many of the available synthetic antidiabetic drugs are not tolerated by patients due to their various side effects [107]. These side effects may include weight gain, hypoglycemia [108], gastrointestinal problems [109], and increased risk of pancreatitis [110]. Thus, there is an urgent need to develop alternative antidiabetic drugs from natural sources, particularly from food.

One of the novel approaches for managing diabetes, particularly type II diabetes, which is the most common, is the inhibition of dipeptidyl peptidase IV (DPP-IV) enzyme. This inactivates glucagon-like peptide-1 (GLP-1), an endogenous incretin hormone, which lowers the secretion of glucagon and increases the secretion of insulin, and in doing so, reduces glucose levels [111,112]. Most potent antidiabetic synthetic drugs have been reported to inhibit DPP-IV enzymes for the management of blood sugar level [113]. Several bioactive compounds, particularly BPs, also show promising antidiabetic properties. Different food proteins have been utilized to generate peptides having DPP-IV-inhibitory properties as summarized in Table 3. Diprotin A (isoleucine–proline–proline; IPP), considered as the most potent dipeptidyl peptidase inhibitor [114], has been isolated from food proteins including chicken egg ovotransferrin and bovine milk casein (ĸ-casein) [6]. BPs isolated from camel milk using bromelain and alcalase also demonstrated DPP-IV-inhibitory potentials [115]. Amino acid sequence ATNPLF, FEELN, AKSPLF, and LSVSVL BPs isolated from black bean protein had strong antidiabetic properties [93]. In the same study, there was a 24.5% reduction in postprandial glucose in rats fed with 50 mg BPs/body weight, when the oral glucose tolerance test was taken. Similarly, GPAE and GPGA isolated from Atlantic salmon skin using flavourzyme at a 6% enzyme/substrate ratio showed potent inhibition of DPP-IV with IC_50_ value of 1.35 mg/mL for peptide fractions below 1 kDa [50]. Nongonierma, Paolella, Mudgil, Maqsood, and FitzGerald [6] identified two potent DPP-IV-inhibitory peptides, LPVPQ and WK, from camel milk protein prepared using trypsin, which possessed a high IC_50_ value (0.68 mg/mL). This value is comparable to that isolated from bovine milk protein hydrolysate (0.85 mg/mL).

### 3.4. Antiproliferative Properties of BPs

In economically developed and developing countries, cancer remains the first and second, respectively, leading cause of death [116]. BPs could act as an antiproliferative substance, which retard or prevent the spread of cells, particularly malignant cells, into surrounding tissues. Induction of apoptosis, immunostimulation, promotion of cell cycle arrest, attenuation of tumor angiogenesis, inhibition of tumor cell-mediated protease activity, or radical scavenging activities are possible mechanisms for suppressing tumor genesis by BPs [116].

Molecular weight, amino acid composition, hydrophobicity, and concentration are key factors affecting the antiproliferative properties of food-derived BPs. The antiproliferative properties of BPs obtained from tuna dark muscle by-product using protease XXIII and papain appear to be dependent on the size of peptides. Fractions with amino acid sequence Pro–Thr–Ala–Glu–Gly–Gly–Val–Tyr–Met–Val–Thr and Leu–Pro–His–Val–Leu–Thr–Pro–Glu–Ala–Gly–Ala–Thr having molecular weights of 1124 and 1206 Da, respectively, possessed the highest antiproliferative activity against human breast cancer cell line MCF-7 (IC_50_ values of 8.8 and 8.1 μM, respectively) [117]. The purified BP fractions rich in Glu–Arg–Asp–Glu, which were obtained from hydrolysate isolated from Nemipterus japonicus backbone using trypsin, had antiproliferative activity against a human hepatoblastoma cell line (Hep G2), with an IC_50_ value of 61.1 μg/mL [118]. Hydrolysate fractions obtained from Indian salmon using trypsin and pepsin with low molecular weight (<5 KDa) showed higher cytotoxicity and antiproliferative activity against human breast cancer MCF-7 cell lines at higher concentrations (10 and 20 µg/mL) as compared to fractions with a higher molecular weight of 5–10 and >10 kDa [119]. A study performed on the antiproliferative activity of peptides isolated from fish by-products (bones, skin, head, gills) on human colon (COLO320) and breast cancer (MCF7A) cells revealed that BPs exerted growth inhibition on the cancer cells at 1 g/L [120]. In particular, the BPs prepared from gills and skin showed growth inhibition of 28.5% and 47.1%, respectively, after 48 h, and the values recorded were in the same range as etoposide, a reference anticancer molecule.

Regarding plant protein-derived BPs, the antiproliferative properties of protein hydrolysate from soybean and mungbean have been examined. Peptide fraction >10 kDa produced from pepsin and pancreatin hydrolyzed soy protein hydrolysate showed the highest antiproliferative activity with CasKi (IC_50_ = 14.3 mg/mL) and HeLa (IC_50_ = 16.2 mg/mL) cervical cancer cells. The peptide fraction also showed the greatest sensitivity (IC_50_ = 15.2 mg/mL) in MDA-MB-231 breast cancer cells [98] Mungbean BPs prepared using papain inhibited the proliferation of the tumor cells in mice and promoted apoptosis of HepG2 cells and arrested the cell cycle in the S phase at a low dose and G0/G1 phase at a high dose [97].

### 3.5. Antimicrobial Properties of BPs

Antimicrobial peptides can exert a wide range of effects (viability and growth) on viruses, bacteria, and yeast, making them promising alternatives to synthetic antimicrobials [121]. Studies have shown that the presence of hydrophobic and hydrophilic amino acids at the terminal is what enables antimicrobial peptides to interact with microbes [36]. This interaction allows the peptides to destroy bacteria either through interaction with various macromolecules within microbial cells or by perforating the bacterial cell through the cell membrane to disrupt the cellular activity and release the cell content [122,123]. The mechanism of action of antimicrobial peptides on bacterial cells has been previously investigated [124].

Numerous BPs with promising antimicrobial activity have been isolated from foods over the years [123,125]. Potent antimicrobial BPs (ELLLNPTHQIYPVTQPLAPV) generated from human colostrum using disk diffusion and flow cytometry methods inhibited *Staphylococcus aureus* and *Yersinia enterocolitica* cells through the destruction of cytoplasmic membrane and cell wall [123]. Additionally, antimicrobial BPs isolated from αs2-casein could inhibit *E. coli*, *Micrococcus luteus*, *Listeria innocua*, and *Salmonella enteritidis*. Potent antimicrobial peptides may also be generated from other sources, including mackerel by-product [126], yellowfin tuna [127], and fish skin hydrolysate [128].

In summary, BPs derived from food proteins have shown promising potential for the management of chronic and acute diseases. From a techno-functionality perspective, there is a need to explore several techniques for producing BPs with high yield and improved bioactivities. The following sections provide an insight into the intensification of production of BPs generated from food proteins using ultrasonication, high-pressure processing, and pulsed electric field (PEF), which are emerging green processing technologies. The mechanism of conformational changes in protein structure and the impact of these non-thermal technologies on the bioactivities of the generated BPs are discussed in the following sections.

## 4. Non-Thermal Pre-Treatments and Their Impacts on Yield and Bioactivities of BPs

Thermal treatment has been applied to food proteins before hydrolysis to enhance the hydrolytic reaction owing to the increase in susceptibility of peptide bonds to enzyme cleavage as well as inactivating indigenous enzymes [26]. However, alterations in structural configurations, particularly in the secondary structure of native proteins, usually occur when subjected to heat treatment [129]. This change might negatively influence the yield and bioactivities of BPs produced. Liu et al. [130] revealed that changes in the secondary structure of lactoferrin mediated by heat resulted in a conformational shift and molecular unfolding. Heat treatment could lead to thermal oxidation in fish muscle, which might decrease bioactivities of the resulting BPs [131]. In addition, some enzymes have high heat resistance and can only be inactivated when high temperature and longer treatment times are employed, which could negatively impact the quality of food proteins [132]. Hence, an alternative pre-treatment technique as an alternative to heat is required to maximize BP production. Ultrasonication, pulsed electric field, and high pressure are the common non-thermal processing technologies that can be applied prior to enzyme hydrolysis to enhance the yield and bioactivities of BPs generated from different food proteins [9,28,33,130,133,134]. A summary of the impact of various non-thermal technologies on bioactivity of peptides derived from food proteins is presented in Table 4.

### 4.1. Ultrasonication

Ultrasonication is a technique that utilizes the frequency of sound waves above 20 kHz; a frequency higher than the detection level of human audition [3]. Ultrasound generally acts by generating acoustic cavitation in the biological matrix, and it has been found to have various effects on the bioactivity and functionality of food proteins. An illustration of how the ultrasound process could modify protein structure and enhance bioactivity is presented in Figure 3. Ultrasound waves (UWs) carry high energy while traveling through a medium [140]. Alternatively, UWs compress and stretch during their movement in a medium; thereby, the build-up of localized negative pressure is formed. When the stretching phase or rarefaction occurs, the generated pressure is sufficient to overcome intermolecular binding forces, which results in the creation of cavitational bubbles or tiny cavities in the medium. This phenomenon is termed as cavitation [141]. These bubbles or cavities continue to grow bigger with successive cycles and later collapse violently; thus, a huge amount of energy is released into the system. The localized pressure can be increased to more than 400 MPa and when the bubbles collapse, it can disrupt covalent and disulfite bonds in protein, thereby promoting enzymolysis [3]. Sonication can enhance enzymatic hydrolysis of food proteins via sonochemical reactions, microstreaming, sonolysis of water, and the formation and collapse of cavitational bubbles [142]. In addition, low and high-frequency ultrasounds are also employed in peptide drug delivery applications, and their effectiveness has been documented [143,144]. The applications of ultrasound in food processing and preservation have been thoroughly described elsewhere [145,146].

#### 4.1.1. Effect of Ultrasound on the Structure of Food Proteins

It is known that proteins possess four levels of structure (e.g., primary, secondary, tertiary, and quaternary structures) [147], which the application of ultrasound can alter [148,149,150,151]. Generally, changes in the structural configuration of proteins following ultrasound treatment may be due to the ability of ultrasound energy to disrupt the intermolecular interactions between protein chains as well as disulfide bonds in proteins [152]. The impact of ultrasound on the protein structure can be attributed to extremely (localized) high temperature (above 1000 °C) and pressures (ranging from 50 to 500 MPa) generated by cavitation, which occurs when food samples are subjected to ultrasound treatment [132,153]. This extremely high acoustic energy generated during ultrasound treatment can disrupt the non-covalent bonds (e.g., hydrogen bonds) and disulfide bonds in proteins. These disruptions play a vital role in the conformational changes in the secondary and tertiary structures of proteins [29], allowing enzymes to access the loosened structure quickly, thereby enhancing the rate of hydrolysis and degree of bioactivity [154].

Several studies have shown ultrasound to be useful in modifying the structural and functional properties of food proteins [150,155,156,157,158]. Zhao, Liu, Ding, Dong, and Wang [158] studied the impact of high-intensity ultrasound (20 kHz frequency; power from 600 to 2000 W; time from 0–30 min) on the structural properties of soy protein isolate (SPI) using Fourier-transform infrared spectroscopy (FTIR), near-infrared spectroscopy (NIR), and fluorescence spectra analyses. They reported that ultrasound treatment altered the secondary structure of the proteins by decreasing the proportion of β-turns and α-helices and increasing the content of β-sheets and random coils. In addition, structural changes in tertiary structure due to the impact of ultrasound treatment was revealed by fluorescence spectra analysis. Xue et al. [159] also reported a reduction in α-helix and an increase in random coils in ultrasound-treated (ultrasonic intensity of 544.59 W/m^2^ for 80 min at 70 °C) buckwheat protein isolate in comparison to untreated samples. In another study, changes in the secondary structure of almond milk proteins were investigated using circular dichroism (CD) and FTIR. FTIR spectra indicated that ultrasound treatment caused minor relocation in the structural configuration of the proteins. Additionally, ultrasound treatment (20 kHz frequency for 16 min at 50% duty cycle duration) caused a reorganization of β-sheets and α-helices as revealed by CD spectroscopy [160]. Furthermore, Fadimu, Gill, Farahnaky, and Truong [29] studied the impact of ultrasonic pre-treatment on structural properties of lupin protein using FTIR, XRD, and circular dichroism. They observed that ultrasound treatment altered the secondary structure of the proteins.

#### 4.1.2. Effect of Ultrasound Pre-Treatment on the Bioactivities of BPs Produced Using Enzymolysis

*ACE-inhibitory properties.* In general, contradictory findings were reported regarding the application of ultrasound pre-treatment on the ACE-inhibitory property of food proteins. It seems that the effects of ultrasound on the bioactivity of BPs are dependent on the enzyme type, type of protein used, sonication time, and frequency applied, which in turn influences the MW of peptide produced. For instance, the potential of ultrasound for enhancing the bioactivity of cheddar cheese during processing has been documented [161]. In the study, cheddar cheese subjected to ultrasound treatment (at 20 kHz frequency and specific energy 41 J/g) prior to ripening showed higher ACE-inhibitory activity and lower IC_50_ value than those of control counterparts. Jia et al. [162] reported that ultrasound treatment during enzymatic hydrolysis of wheat germ protein using alcalase (6625.4 U) for 210 min had less effect on ACE-inhibitory activity of the hydrolysate. On the contrary, it was revealed that the ACE-inhibitory property of hydrolysates prepared using bromelain increased by 13% (from 51.10% to 72.13%) after sonication for 5 min at ultrasound density of 0.092 W/mL [31]. Nevertheless, sonication did not improve the ACE-inhibitory property of papain-hydrolyzed whey protein [33]. This enzyme specificity is also documented in an earlier study by Uluko, Zhang, Liu, Chen, Sun, Su, Li, Cui, and Lv [134] where sonication was applied to milk protein hydrolysis. Dabbour et al. [163] reported that significant ACE-inhibitory activity (64.46%) was obtained in sunflower meal protein subjected to ultrasound treatment (20/40 kHz, 15 min, 220 W) prior to enzymolysis compared to the control sample (32.52%). In a related study, Fadimu et al. [30] reported a significant increase in ACE-inhibitory activity of lupin protein hydrolysate pre-treated using ultrasound in comparison to unsonicated samples.

*Antioxidant activity.* Ultrasonication appears to be effective in enhancing the antioxidant activity of BPs in most cases. The improvement of antioxidant properties of BPs obtained from plant protein via ultrasound pre-treatment have been documented. Sonication before enzymolysis (alcalase at 0.01 Au/g E/S ratio for 30 min) in a frequency range of 25–69 kHz tended to intensify the hydrolysis of wheat gluten, with a sample treated at the lowest frequency (25 kHz) exhibiting the lowest EC_50_ value of 0.279 mg/mL. This indicates higher ferrous ion chelating activity in comparison to samples obtained using a high-frequency treatment which had an EC_50_ value of 0.513 mg/mL [34]. In addition, the antioxidant activity of ultrasound treated papain-hydrolyzed (1:20 E/S ratio for 180 min) whey protein increased from 19.62% to 21.17%. However, sonication did not have any effect on sonicated whey protein hydrolyzed when bromelain was used for hydrolysis [27]. In another study, Fadimu et al. [32] revealed that the IC_50_ value of lupin protein hydrolysate was lower (3.77 mg/mL) in ultrasonicated samples in comparison to unsonicated hydrolysate (4.30 mg/mL).

In addition to plant protein, animal protein has been pre-treated with ultrasound for the improvement of the antioxidant properties of the BPs obtained via enzymatic reaction. Jovanović et al. [164] revealed that ultrasound treatment of egg white protein hydrolysate at 40 kHz frequency for 15 min yielded hydrolysate with higher DPPH (28.10%), ABTS (79.44%), and FRAP (0.097 µM/mg) than untreated hydrolysates, which had 15.8%, 7.69%, and 0.062 µM/mg for DPPH, ABTS, and FRAP, respectively. Ultrasound treatment accelerated enzymatic hydrolysis and increased yield of soluble proteins and antioxidant activity (from 19.62% to 22.31%) of hydrolysate prepared from vegetable protease-fermented whey protein [33].

To further increase the bioactivities of BPs produced from food protein, ultrasound has been combined with heat treatment (thermosonication). A combination of ultrasound (60% amplitude, 20 kHz frequency for 10 min) and heat (95 °C) enhanced the antioxidant activity of hydrolysates from duck egg albumen [9]. Despite all these advantages, the application of ultrasound as a pre-treatment method for food protein prior to enzymolysis is still in its primary state. Therefore, it is imperative for researchers to be directed towards the optimization of ultrasound process parameters for the pre-treatment of food proteins.

### 4.2. High-Pressure Processing (HPP)

HPP, referred to as high hydrostatic pressure processing or ultra-high pressure (UHP), involves the exposure of food materials to pressure (100 to 1000 MPa) where instant and even transmissions of pressure throughout the sample allow inactivation of microorganisms without significant changes in quality attributes and nutritional components [165]. As illustrated in Figure 4, typical HPP equipment components include a pressure vessel, high-pressure intensifier pumps, closures, and a device (e.g., yoke) to secure the pressure vessel during processing. Common pressure-transmitting fluids used in food industry are water, castor oil, ethanol, and glycol. The HPP technique has been successfully exploited to meet diverse consumer demands by generating novel foods, tastes, and textures in the last 20 years [26,165]. Lou et al. [166] and Abera [167] provided updated reviews of working principles and applications of HPP in food systems. Apart from the major applications such as improving food safety and stability, freezing, thawing, and extraction, HPP has also been used in combination with protease treatments to produce BPs from food proteins and enhance their bioactivity [37].

#### 4.2.1. Effect of HPP on the Structure of Food Proteins

As aforementioned, proteins are stabilized in their native state by covalent bonds (including disulfide bridges), electrostatic interactions (ion pairs, polar groups), hydrogen bridges, and hydrophobic interactions. The covalent bonds in proteins are usually not affected by HPP, especially at low temperatures (0–40 °C); hence, the primary structure of proteins is intact during HPP treatment [168]. However, HPP treatments affect the secondary structure via hydrogen ion and electrostatic interactions, the tertiary structure via hydrophobic and hydrogen bonding, as well as the quaternary structure through hydrophobic interactions [168]. At a pressure <300 MPa, these changes in protein structures are reversible, but at pressures >300 MPa, an irreversible denaturation of proteins can occur. At pressure above 700 MPa, HPP may give rise to irreversible protein denaturation by disrupting the secondary structure of the proteins [169]. The conformation changes induced by HPP in food proteins can expose the active site for the proteolytic enzymes to effectively cleave the proteins. As such, HPP can be beneficial in improving the generation of bioactive peptides, as illustrated in Figure 4.

In another aspect, it is shown that HPP can inactivate indigenous enzymes in food protein prior to enzymolysis. The native state of an enzyme is stabilized by different types of interactions such as covalent bonding, hydrogen bonding, van der Waals, hydrophobic, and electrostatic interactions [170], which are susceptible to HPP treatment. HPP inactivation of enzymes involves the formation and/or disruption of numerous interactions and change in the native structure of enzymes by folding and/or unfolding, which could be reversible or irreversible depending on the applied pressure [171]. Consequently, this will lead to a change in enzyme activity as its specificity is related to the enzyme’s active site structure. Thus, the denaturation of enzymes, similar to the protein denaturation under the HPP environment, is the main inactivation mechanism for HPP for enzymes [172].

#### 4.2.2. Effect of HPP Pre-Treatment on the Bioactivity of BPs Produced Using Enzymolysis

Although the bioactivities of BPs mainly depend on the properties of the BPs, the pre-treatment method prior to enzymolysis may have a significant effect on the properties of the produced BPs. The efficacy of HPP on the bioactivities of BPs from different commodities varies. Generally, it has been reported that HPP affects the bioactivities of peptides from food protein via the increased susceptibility of proteins to digestion during proteolysis [173]. HPP pre-treatment at 300 MPa on lentils prior to hydrolysis using savinase enzyme produced BPs with high ACE-inhibitory and antioxidant activities when compared to the BPs produced with the same enzymes without prior pre-treatment [28]. Quirós et al. [174] reported that pre-treatment of ovalbumin at a pressure range between 200 and 400 MPa facilitated the release of novel peptides RADHPFL, YAEERYPIL, and FRADHPFL, which showed antihypertensive activity in vitro. In another study, HPP treatment of lentil protein at a pressure above atmospheric pressure (0.1 MPa) enhanced the oxygen radical absorbance capacity (ORAC) (from 36.6 to 70.77%) and ACE-inhibitory activity (from 245.16 to 360.99 µmoles TE/g) of BPs produced using savinase, corolase, and protamex [28]. According to Zhang, Jiang, Miao, Mu, and Li [38], application of HPP in a lower pressure range (between 100 and 200 MPa) to chickpea protein and hydrolysis using alcalase increased the superoxide anion capturing rate and caused a reduction in reducing power of the hydrolysates from 27.26% to 66.26% and 0.134 to 0.406, respectively. Even though HPP treatment in combination with proteases has the potential to enhance the proteolytic efficiency in terms of yield and bioactivities, it was proposed that pressurization above 1000 MPa could cause structural damage and denaturation of protein, which in turn would negatively affect the properties of the BPs produced from food proteins [28]. Hence, the parameters of HPP must be optimized to ensure the improvement of yield and bioactivities of the produced BPs.

### 4.3. Pulsed Electric Field (PEF)

PEF is one of the novel non-thermal technologies that can effectively inactivate microorganisms as well as enzymes in food [175]. PEF is based on electroporation and cell disintegration by applying repeated pulses when the food is placed between two parallel electrodes [176]. In PEF processing, microsecond high-voltage pulses in the range of 10 to 60 kV are employed [177]. The application of high-voltage pulses induces pores in a process named electroporation in cell membranes, initiating a loss of barrier function, intracellular content leakage, and loss of vitality [175]. PEF technology is considered superior to traditional heat treatment of foods based on food-quality attributes because it avoids, or greatly lessens, unfavorable changes in physical and sensory properties [178]. Potentially, PEF can induce conformation changes in enzymes, thereby leading to enzyme inactivation [179]. The activities of protease, lipase, and alkaline phosphatase in fresh bovine milk subjected to PEF treatment at electric field strengths ranging from 15 to 35 kV/cm were reduced to 14%, 37%, and 29%, respectively [180].

#### 4.3.1. Effect of PEF on the Structure of Food Protein

Studies have indicated that non-thermal treatment in the form of electromagnetic wave and electric fields could modify the structure of proteins [181,182] thereby modifying the native structure of protein for optimum delivery of health-promoting peptides. Only secondary and tertiary structures of the protein are affected by the electromagnetic wave and electric fields [133]. The mechanism of the alteration of protein structure may be linked to energy absorption potential by polar groups of proteins and generation of free radicals, which causes unfolding of proteins [133] as well as protein oxidation [183]. Changes in secondary protein structure may be evaluated by qualitatively analyzing the content of α-helices, β-sheets, and β-turns using spectroscopy techniques [184]. On the other hand, changes in the intensity of intrinsic and extrinsic fluorescence may reflect alteration of tertiary structure, which is associated with the presence of aromatic amino acids including phenylalanine and tryptophan for intrinsic fluorescence. However, extrinsic fluorescence may reflect changes in hydrophobicity of proteins [185].

#### 4.3.2. Effect of PEF on the Bioactivities of BPs Produced from Food Proteins Using Enzymolysis

Given that the exact underlying mechanism of PEF in the bioactivity of food protein-derived peptides is not fully understood, it is assumed that changes in protein structure could be responsible for improved bioactivities observed in BPs produced from food protein subjected to PEF before hydrolysis [133,186,187]. According to Lin et al. [188], treatment of egg white protein using high-intensity pulsed electric field at constant parameters of 10 kV/cm electric field strength, 3000 Hz frequency, and pulse number of 300 significantly increased (from 3.0 to 3.5%) the antioxidant activity of the fractionated hydrolysate (<1 kDa) prepared using alcalase in comparison to the untreated sample. This could be due to the breakdown of larger peptides into smaller peptides by breaking down the peptide bonds, electrostatic interactions, hydrophobic interactions, with subsequent disorganization of larger protein structure caused by the PEF pre-treatment [189]. The disorganization could have facilitated the production of smaller peptides with better activities than the larger ones [189].

Lin et al. [137] studied structural and antioxidant activities of a peptide (SHCMN) generated from PEF-treated soybean protein. It was reported that the optimum PEF conditions needed to increase the antioxidant activity of the peptides from 93.43% to 94.35% were 5 kV/cm electric field, 2400 Hz pulse frequency, and retention time of 2 h. The intensification of hydrolysis could be attributed to changes in secondary structure [137]. Apart from those studies performed on the enhancement of antioxidant activities of hydrolysates from food proteins, there is little information about the effects on PEF treatment on other bioactivities including antihypertensive, antidiabetic, antimicrobial, antiviral, opioid, and antithrombotic activities. Thus, there is a need to further explore the utilization of PEF as sole pre-treatment either individually or in combination with other technologies for food proteins prior to enzymolysis to produce BPs with improved yield and bioactivities.

## 5. Conclusions and Remarks

Non-thermal food processing technologies such as US, HPP, and PEF hold tremendously high potential in applications as novel alternatives to heat treatment for enzymatic hydrolysis of food proteins to produce BPs. In general, research has shown that acoustic cavitation, pressure-induced reversible/irreversible change, and high-voltage pulses in US, HPP, and PEF techniques, respectively, can cause conformational changes in food proteins. Pioneering work demonstrated that these primary modes of action contribute to the enhanced susceptibility of protein structures to enzymatic hydrolysis. However, detailed knowledge of underlying mechanisms is still lacking and there remains a need to elucidate the specific mechanisms fully. The process efficiency and bioactive properties of resulting BPs appear to be dependent on various factors, including US, HPP, PEF operating conditions, types and concentration of enzymes used, pH, and temperature. This prompts the quest for optimization of the process parameters to deliver specific functionality or bioactive properties. In addition, studies performed on plant proteins appear to be less available compared to animal proteins that are also worthy of future investigation.

To the best of our knowledge, intensification of enzymatic hydrolysis by other non-thermal food processing technologies such as cold plasma, dense phase carbon dioxide, radiofrequency electric field, and oscillating magnetic fields to generate food-derived BPs has not yet been examined. Additionally, the combination of different non-thermal processes may provide a synergistic effect in improving the pre-treatment efficiency. Thus, the utilization of non-thermal food processing technologies will provide a robust approach and competitive alternatives to the conventional thermal methods as pre-treatment in enzymatic hydrolysis of food proteins. From a scientific point of view, full elucidation of mechanisms by which non-thermal food processing technologies in combination with enzyme treatment cause the structural changes in protein is pivotal to understanding how bioactivity is enhanced.

## Figures and Tables

**Figure 1 foods-11-01823-f001:**
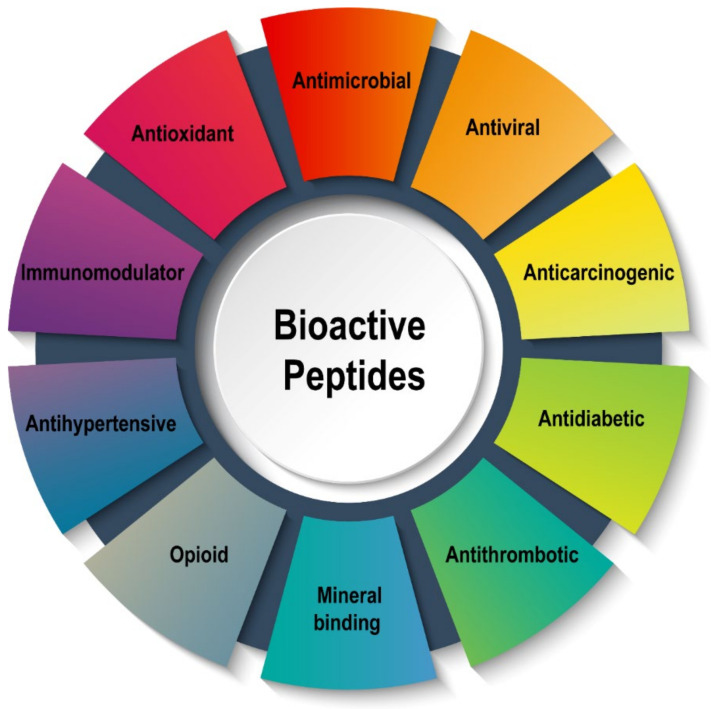
Schematic representation of various types of bioactive peptides from food proteins.

**Figure 2 foods-11-01823-f002:**
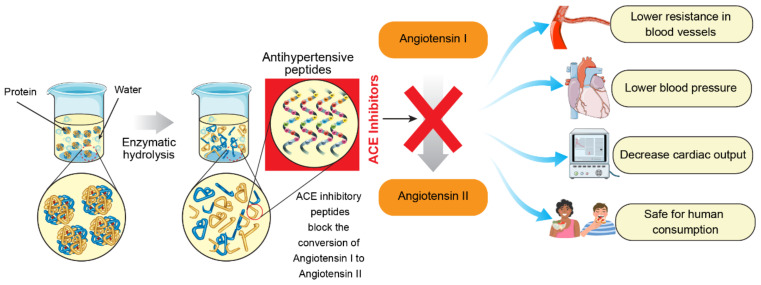
Schematic illustration of preparation and mechanism of action of ACE-inhibitory peptides.

**Figure 3 foods-11-01823-f003:**
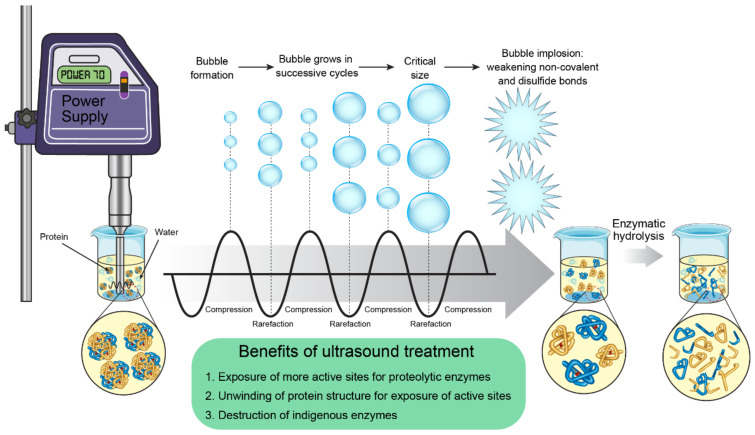
Schematic illustration of ultrasound-assisted proteolysis of food proteins.

**Figure 4 foods-11-01823-f004:**
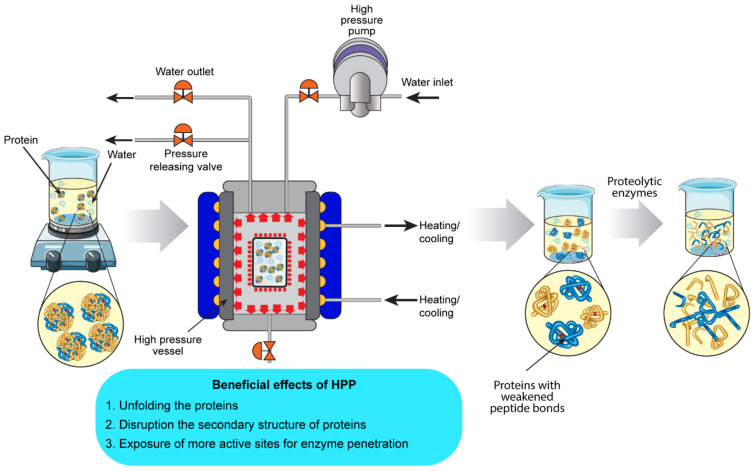
Schematic illustration of HPP-assisted proteolysis of food proteins.

**Table 1 foods-11-01823-t001:** Proteases commonly used for the production of peptide-rich hydrolysates.

Proteases	References
Flavourzyme	[22,23]
Trypsin	[23,42]
Chymotrypsin	[43]
Alcalase	[13,22,43,44,45]
Flavorase	[45]
Papain	[15,45]
Protamex	[30,46]
Pepsin	[15,23,47,48]
Neutrase	[43]
Bromelain	[33,49,50]
Corolase	[28,51]
Neutral protease	[52]
Savinase	[28]
Thermolysin	[53]
Protease P	[54]

**Table 2 foods-11-01823-t002:** Antioxidant peptides from food proteins.

Group	Source	Method of Preparation	Amino Acid Sequence	Major Findings	References
Plant protein	Corn protein	Hydrolysis using alcalase	SGV, LLPH, NGGGA	Peptides identified in the study showed strong antioxidant activity	Liang, Ren, Ma, Li, Xu, and Oladejo [72]
	Chickpea albumin	Hydrolysis using flavourzyme and alcalase	RQSHFANAQP	Strongest antioxidative peptides were identified in the fractionated chickpea hydrolysate	Kou, Gao, Xue, Zhang, Wang, and Wang [22]
	Coconut cake albumin hydrolysate	Hydrolysis using alcalase, trypsin, pepsin, and flavourzyme	KAQYPYV, KIIIYN, KILIYG	Identified peptides showed strong ion chelating ability and substantial superoxide radical scavenging activity	Zheng, Li, and Li [23]
	Rice endosperm protein	Hydrolysis using alcalase, neutrase, papain, flavorase, and chymotrypsin	FRDEHKK, KHDRGDEF	Most potent antioxidant peptides were released by neutrase	Zhang et al. [45]
Animal protein	Goat milk casein	Hydrolysis using pepsin	TVNREQL, VNQELAYFYPQLFRQ, DMESTEVF, QSLVYPFTGPI	Protein hydrolysates showed strong antioxidant activity. Four potent antioxidant peptides were identified	Ahmed et al. [47]
	Buffalo milk	Hydrolysis using papain, pepsin, and trypsin	KFQ, YPSG, HPFA	Antioxidant peptides were identified in papain hydrolysates	Abdel-Hamid, Otte, De Gobba, Osman, and Hamad [15]
	Buffalo casein	Hydrolysis using alcalase and trypsin	RELEE, TVA, MEDNKQ, EQL	Peptides with molecular weight below 1 KDa exhibited strongest antioxidant activity	Shazly et al. [43]
	Egg white protein	Hydrolysis using alcalase	FFGFN, DHTKE, MPDAHL	Out of the three identified peptides, DHTKE showed highest ORAC	Liu et al. [73]
	Chicken egg white	Hydrolysis using protease P	DEDTQAMP, AEERYP	Only two of the sixteen identified peptides showed very strong oxygen radical absorbance capacity (ORAC)	Nimalaratne et al. [54]
	Camel milk	Hydrolysis using papain, trypsin, alcalase, and pepsin	RLDGQGRPRVWLGR, TPDNIDIWLGGIAEPQVKR, VAYSDDGENWTEYRDQGAVEGK	Peptides showed strong DPPH and ABTS activities	Wali et al. [48]
Fish and fish products	Seabass skin	Hydrolysis using alcalase	GLPGPA, GATGPGGPLGPA, VLGPP, GLGPLGPV	Hydrolysates showed strong DPPH radical scavenging activity	Sae-leaw et al. [67]
	Grass carp	Hydrolysis using alcalase	PYSFK, GFGPEL, GGRP	Three isolated peptides exhibited high ABTS, DPPH, and hydroxyl radical activities in a dose-dependent manner	Cai et al. [74]
	Unicorn leatherjacket	-	GPGPVG, LPGPAG, LAGPVG, GGPLG	Isolated peptides showed cellular antioxidant activity by protecting H_2_O_2_-induced DNA damage	Karnjanapratum et al. [66]
	Croceine croaker muscle	Hydrolysis using alcalase	VLYEE, YLMSR, MILMR	YLMSR exhibited highest DPPH, superoxide, and ABTS activity	Chi et al. [75]
	Carp skin gelatin	Hydrolysis using protamex	AY	Peptide identified is responsible for high antioxidant activity recorded	Tkaczewska et al. [76]
	Pacific cod	Hydrolysis using trypsin	AGPAGPAGAR, GPAGPHGPPGKDGR, AGPHGPPGKDGR	Identified peptide exhibited high iron chelating activity	Wu et al. [77]
	Bluefin leatherjacket (*Navodon septentrionalis*)	Hydrolysis using alcalase, neutrase, and flavourzyme	FIGP, GPGGFI, GSGGL	All identified peptides contributed to the DPPH and hydroxyl radical scavenging activity of the hydrolysate	Chi et al. [78]
	Amur sturgeon	Hydrolysis using alcalase and flavourzyme	PAGT	The peptide exhibited DPPH, ABTS, and hydroxyl radical scavenging abilities	Nikoo et al. [52]

**Table 3 foods-11-01823-t003:** ACE-inhibitory, antimicrobial, and antiproliferative peptides from food proteins.

Biological Activity	Food Source	Method of Preparation	Peptide Sequence	Major Findings	References
ACE-inhibitory peptides	Fermented camel milk	Fermentation using *L. helveticus* and *L. acidophilus*	LSLSQF, KVLVPQ, FQEPVPDPVR, LENLHLPLPL, KVLPVPQQMVPYPQ, VMVPFLQPK	Seven ACE-inhibitory peptides identified in *L. helveticus* sample and 3 from *L. acidophilus* sample	Alhaj [90]
	Sesame protein powder	Enzymatic hydrolysis using thermolysin	LVY, LSA, LKY, IVY, MLPAY	Identified peptides had strong antihypertensive effect on spontaneously hypertensive rats	Nakano, Ogura, Miyakoshi, ISHII, Kawanishi, Kurumazuka, Kwak, Ikemura, Takaoka, and Moriguchi [53]
	Mung bean protein	Enzymatic hydrolysis using bromelain	LRLESF, LPRL, HLNVVHEN, YADLVF, PGSGCAGTDL	Five ACE-inhibitory peptides identified from molecular weight fraction below 1 KDa and LRLESF was the most potent peptide	Sonklin, Alashi, Laohakunjit, Kerdchoechuen, and Aluko [49]
	Salmon protein	Enzymatic hydrolysis using corolase	IWHHT, ALPHA, IVY, VW, VPW, TVY, IW, IY	Eleven ACE-inhibitory peptides identified from in vitro, ex vivo, and in silico studies of salmon protein hydrolysate	Darewicz, Borawska, Vegarud, Minkiewicz, and Iwaniak [51]
	Rice bran protein	Enzymatic fermentation using protease G6	GSGYF	Identified peptide showed strong ACE-inhibitory activity with IC_50_ value close to that of captopril	Suwannapan, Wachirattanapongmetee, Thawornchinsombut, and Katekaew [91]
	Buffalo milk	Enzymatic hydrolysis using pepsin, papain, and trypsin	IPPK, QPPQ, FPGPIPK, IVPN	ACE-inhibitory peptides were identified only in papain hydrolysates	Abdel-Hamid, Otte, De Gobba, Osman, and Hamad [15]
	Coconut cake albumin hydrolysate	Enzymatic hydrolysis using alcalase, trypsin, and pepsin	KAQYPYV, KIIIYN, KILIYG	Identified peptides showed strong ACE-inhibitory potential and KAQYPYV was stable against gastrointestinal digestion enzymes	Zheng, Li, and Li [23]
	Goat milk	Enzymatic hydrolysis using alcalase	SLPQ, TGPIPN, SQPK	ACE-inhibitory peptides identified had high IC_50_ values and TGPIPN passed monolayer of Caco-2 cells intact in small quantity	Geerlings, Villar, Zarco, Sánchez, Vera, Gomez, Boza, and Duarte [92]
Antidiabetic peptides	Camel milk protein	Enzymatic hydrolysis using trypsin	LPVPQWK	Potent and unique peptide with DPP-IV-inhibitory activity was identified in camel milk protein hydrolysate for the first time	Nongonierma, Paolella, Mudgil, Maqsood, and FitzGerald [6]
	Black bean	Enzymatic hydrolysis using alcalase	AKSPLF, ATNPLF, FEELN, LSVSVL	Protein hydrolysate from black bean caused a reduction in blood glucose of hyperglycemic rat	Mojica, de Mejia, Granados-Silvestre, and Menjivar [93]
	Atlantic salmon skin gelatin	Enzymatic hydrolysis using bromelain, flavourzyme, and alcalase	GPGA, GPAE	Flavourzyme hydrolysate at 6% enzyme substrate ratio showed higher dipeptidyl peptidase activity than alcalase and bromelain hydrolysate	Li-Chan, Hunag, Jao, Ho, and Hsu [50]
	Camel milk protein	Enzymatic hydrolysis using trypsin	LPVP, MPVQA	Nine novel DPP-IV peptides were identified in the hydrolysate and the most potent two had IC_50_ values comparable to that of pure peptides	Nongonierma, Paolella, Mudgil, Maqsood, and FitzGerald [12]
	Goat milk casein	Enzymatic hydrolysis using trypsin	AWPQYL, SPTVMFPPQSVL, MHQPPQPL, VMFPPQSVL, INNQFLPYPY	Five new peptides with DPP-IV-inhibitory activity were identified and isolated using 2D-TLC. One of the peptides (INNQFLPYPY) showed remarkable IC_50_	Zhang et al. [46]
	Walnut protein	Enzymatic hydrolysis using alcalase	LPLLR	The identified peptide showed strong inhibitory action against α-glucosidase and α-amylase	Wang, Wu, Fang, Liu, Liu, Li, Shi, Li, and Min [13]
Antimicrobial peptides	Mackerel hydrolysate	Enzymatic hydrolysis using protamex	SIFIQRFTT	All four identified peptides partially inhibited Gram positive (Listeria innocua) and Gram negative (Escherichia coli) bacterial strains while SIFIQRFTT totally inhibited both strains	Ennaas et al. [94]
	Anchovy cooking waste	Enzymatic hydrolysis using protamex	GLSRLFTALK	Identified peptide showed no hemolytic activity and exhibited bactericidal effect in reconstituted milk	Tang et al. [95]
	Whey protein	Enzymatic hydrolysis using pepsin, chymotrypsin, and trypsin	VRT, KVGIN, PGDL, KVAGT, EKF, LPMH	Trypsin and chymotryptic hydrolysates did not exhibit antibacterial activity; only hydrolysate from pepsin showed significant activity	Théolier et al. [96]
Antiproliferative peptides	Mung bean protein	Enzymatic hydrolysis using papain	PQG, LAF, EGA, VEG	Identified peptides exhibited in vitro and in vivo anticancer activities	Li, Zhang, Xia, and Ding [97]
	Chickpea protein	Enzymatic hydrolysis using trypsin and pepsin	RQSHFANAQP	Identified peptide inhibited breast cancer cells	Xue et al. [42]
	Germinated soybean	Enzymatic hydrolysis using pepsin	-	Inhibited cervical and breast cancer cells	Marcela, Eva, Del Carmen, and Rosalva [98]

**Table 4 foods-11-01823-t004:** Impact of different non-thermal treatments on biological activities of peptides derived from food proteins.

Source of Peptides	Non-Thermal Treatment	Enzyme Used for Preparation	Major Findings	Reference
Lentil protein hydrolysate	High-pressure processing (HPP); 100 to 300 MPa at 40 °C for 15 min	Alcalase, protamex, savinase, corolase	HPP increased ACE-inhibitory activity of hydrolysates from all the enzymes when compared with control, with exception of alcalase In comparison to control, HPP increased oxygen radical absorbance capacity of all hydrolysate samples	Garcia-Mora, Peñas, Frias, Gomez, and Martinez-Villaluenga [28]
Lupin protein hydrolysate	Ultrasound (ultrasonic power: 400 W, frequency: 20 kHz, time: 10 min)	Flavourzyme (pH: 6.0 at 60 °C), alcalase (pH: 8.0 at 50 °C), protamex (pH: 8.0 at 50 °C) for 4 h	Ultrasound increased antioxidant, antihypertensive, α-amylase, and α-glucosidase activities when compared with control	Fadimu, Gill, Farahnaky and Truong [31], Fadimu, Gill, Farahnaky, and Truong [32], Fadimu, Farahnaky, Gill, and Truong [30]
Peanut protein hydrolysate	Ultrasound (ultrasonic power: 150 W, time: 25 min)	Alcalase (pH: 8.5 at 60 °C)	DPPH radical scavenging activity increased up to 90% after ultrasonic treatment	Yu et al. [29]
Wheat gluten hydrolysate	Ultrasound (ultrasonic frequency: 25–69 kHz, ultrasound intensity: 0.707 W/cm^2^)	Alcalase (pH: 9.0; temperature: 50 °C for 30 min)	Hydrolysate obtained under ultrasound treatment exhibited highest iron chelating and reducing power in a dose-dependent manner	Zhu, Su, Guo, Peng, and Zhou [34]
Whey protein hydrolysate	Ultrasound (ultrasound density: 0.092 W/mL, time: 5 min)	Bromelain (pH 7.0 at 50 °C); papain (pH 7.0 at 60 °C for 180 min)	ACE-inhibitory activity increased from 13 to 95% in bromelain hydrolysate, but did not improve in papain hydrolysate Ultrasound treatment did not improve antioxidant activity	Abadía-García, Castaño-Tostado, Ozimek, Romero-Gómez, Ozuna, and Amaya-Llano [33]
Duck albumen hydrolysate	Ultrasound (amplitude: 60%, time: 10 min, power: 750 W)	Papain, alcalase (pH 8.0 at 50 °C for 4 h)	Ultrasound pre-treatment improved the antioxidant activity after 90 min of hydrolysis	Quan and Benjakul [9]
Corn protein hydrolysate	Ultrasound (frequency: 28 kHz, time: 25 min, power: 65 W/L)	Alcalase (pH 9.0 at 50 °C)	Sonication treatment increased antioxidant activity from 60 to 65%	Liang, Ren, Ma, Li, Xu, and Oladejo [72]
Erythrina edulis hydrolysate	Ultrasound (amplitude: 100%, time: 10 min, frequency: 80 kHz)	Flavourzyme, alcalase	DPPH and ABTS radical scavenging activity significantly increased following ultrasonic pre-treatment in comparison to untreated hydrolysates ACE-inhibitory activity increased in a dose-dependent manner after ultrasonic treatment	Guerra-Almonacid et al. [135]
Rapeseed protein hydrolysate	Ultrasound (power: 600 W, time: 12 min)	Alcalase (pH 9.0 at 50 °C for 120 min)	Ultrasonic treatment increased ACE inhibitory activity from 51.10 to 72.13%	Wali et al. [136]
Egg white protein	Pulse electric field (electric field: 10 kV/cm, frequency: 3000 Hz, pulse number: 300)	Alcalase, pepsin, trypsin	PEF treatment increased antioxidant activity from 3.0 to 3.5%, with alcalase hydrolysate having highest activity	Lin, Guo, You, Yin, and Liu [137]
Pine nut protein	Pulse electric field (frequency: 1800 Hz, electric field: 15 kV/cm)	-	PEF increased DPPH scavenging activity from 89.10 to 93.22%.	Lin et al. [138]
Caprine milk protein	Ultrasound (power: 200 W, amplitude: 80%, time: 20 min)	Neutral protease (50 °C for 6 h), pepsin (37 °C for 6 h)	Ultrasound pre-treatment caused significant increase in DPPH and ACE-inhibitory activity	Koirala et al. [139]

## Data Availability

Not applicable.

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
