# Peer review of "Enhancing the Biological Activities of Food Protein-Derived Peptides Using Non-Thermal Technologies: A Review"

_foods, 2022, doi:10.3390/foods11131823_

Round 1

Reviewer 1 Report

I reviewed the manuscript entitled, Enhancing the biological activities of food protein-derived peptides using non-thermal technologies: A review. The manuscript is well written; however, the manuscript should be revised before final acceptance.

Add line and page numbers

Abstract: add review results and author recommendations

Introduction: general introduction on US, HPP, and PEF should be added in introduction

Section 2: list or Table or figure representing different enzymes used should be added  

Table 1 is limited to a few sources. Please add more recent studies of peptides from plants, animals, including marine. There are many peptide studies from marine food sources.

Table 2. Please include antidiabetic properties of peptides. It is not mentioned in Table 2.

All scientific names must be in italics throughout the manuscript

Table 3. Column 3. Enzymes used for preparation: enzyme conditions and ph or temperature should be added throughout the Table and try to mention all these conditions, if available in literature. Similarly, for HPP and US, conditions and temperature should be mentioned throughout Table 3.

None of the References from 1 to 191 are according to journal format. Please revise according to journal format. Journal name should be in Italics.

In vitro should be in Italics throughout the manuscript.

Author Response

Reviewer #1:

I reviewed the manuscript entitled, Enhancing the biological activities of food protein-derived peptides using non-thermal technologies: A review. The manuscript is well written; however, the manuscript should be revised before final acceptance.

Question 1: Add line and page numbers.

Response: Thank you for your feedback. We added line and page numbers to the manuscript.

Abstract

Add review results and author recommendations

Response: We appreciate your valuable. We have updated the abstract with information about our major findings and recommendations

Introduction

General introduction on US, HPP, and PEF should be added in introduction

Response: Thank you for your feedback. We have added general information about ultrasound, high-pressure processing, and electric pulse-field to the introduction part of the manuscript.

Section 2: List or Table or figure representing different enzymes used should be added

Response: Thank you for your feedback. We added another table containing a list of proteases commonly used in hydrolysate production.

Table 1 1 is limited to a few sources. Please add more recent studies of peptides from plants, animals, including marine. There are many peptide studies from marine food sources.

Response: Thank you for your insightful comment. Table 1 has been updated with recent information on peptides from plant and animal as suggested.

Table 2 Please include antidiabetic properties of peptides. It is not mentioned in Table 2.

Response. Thank you for your feedback. Information about antidiabetics is available in Table 2 (now Table 3).

All scientific names must be in italics throughout the manuscript

Response: Many thanks for your observation. We italicize all scientific names in the manuscript.

Table 3. Column 3. Enzymes used for preparation: Enzyme conditions and pH or temperature should be added throughout the Table and try to mention all these conditions, if available in literature. Similarly, for HPP and US, conditions and temperature should be mentioned throughout Table3.

Response: Many thanks for your observation. We have provided the pH and temperature for enzymic conditions available in the literature (See Table 4).

None of the references from 1 to 191 are according to journal format. Please revise according to format. Journal name should be in italics.

Response: Thank you for your observation. We have reformatted the references according to journal style.

In vitro should be in italics through the manuscript

Response: We have italicized all in vitro and in silico throughout the manuscript

Reviewer 2 Report

The manuscript describes and discusses logically designed experiments and presents results that are expected to be of large interest to the scientific community. It is an interesting study with an interesting approach. The paper as a whole is well designed and results in sound. Nevertheless, the manuscript needs a major revision:

1. Mention some detailed information in the introduction section regarding the antimicrobial effect produced by peptides. 

Also, In the introduction part should be more highlighted the main aim of the paper, and additionally, what is the novelty of carrying out research work.

2. The quality of the Table must be improved.

3. The manuscript lacks coordination and interrelations between sentences and paragraphs.

Author Response

Reviewer #2:

The manuscript describes and discusses logically designed experiments and presents results that are expected to be of large interest to the scientific community. It is an interesting study with an interesting approach. The paper as a whole is well designed and results in sound. Nevertheless, the manuscript needs a major revision:

  1. Mention some detailed information in the introduction section regarding the antimicrobial effect produced by peptides.

Response: Thank you for your comment. the revised manuscript has been revised following the suggestion of the reviewer. The antimicrobial effect of peptides has been included in the introduction section (Please see line 45-50)

Also, in the introduction part should be more highlighted the main aim of the paper, and additionally, what is the novelty of carrying out research work.

Response: The novelty and aim of the research work has been highlighted in the revised text. Please see line 112-119

  1. The quality of the Table must be improved.

Response: Many thanks for your comment. We have provided further information to improve the quality of the table.

  1. The manuscript lacks coordination and interrelations between sentences and paragraphs.

Response: Thank you for the correction. The revised manuscript has been adjusted accordingly. 

Round 2

Reviewer 1 Report

Authors are now answered the suggestions. This version can be accepted for publication.